# $MC^2$: Multimodal Concept-based Continual Learning

## Abstract

The inability of deep neural networks to learn continually while retaining interpretability limit their deployment in critical settings. Existing research has made strides in either interpretability or continual learning, but the synergy of these two directions largely remains under-explored. This work examines this intersection from the perspective of concept-based models where classes are considered as combinations of text-based concepts, and thus can enhance the interpretability of models in a continual learning setting. Addressing the unique challenges of learning new concepts without forgetting past ones, our method **MC²** proposes an approach to seamlessly learn both classes and concepts over time. We adopt a multimodal approach to concepts, emphasizing text-based human-understandavle semantics associated with images. Through various experimental studies, we demonstrate that **MC²** outperforms existing concept-based approaches by a large margin in a continual setting, while performing comparably if not better in full-data settings. We also demonstrate that $MC^2$ can be used as a post-hoc interpretability method to examine image regions associated with abstract textual concepts. Our code for **MC²** will be publicly released on acceptance.

## 1 Introduction

Modern deep neural networks (DNNs) have proven their ability to solve a multitude of tasks in the supervised learning setting, even outperforming humans on certain tasks. In recent times, there has been growing interest in developing models that not only perform well on a single task in an i.i.d. setting but also *learn continually*, i.e. on new, previously unseen tasks that may arrive in the future. However, in such settings, when a model is directly trained on a new task, it loses the ability to perform well on previously learnt tasks, a phenomenon known as *catastrophic forgetting*. Many methods in continual learning literature have explored a plethora of techniques to combat this issue (Wang et al., 2023). However, models that have the capability to learn continually still have a significant hurdle that prevents them from being extensively deployed in safety-critical conditions - they cannot explain how they arrive at a decision from the provided data, viz., they are not interpretable.

From another perspective, methods that allow deep neural networks to become interpretable have also become an active area of research in recent times (Molnar, 2018; Samek et al., 2021). Most existing literature focus on post-hoc interpretability, i.e. they attempt to explain the decisions of a model already trained on a particular dataset and task. There has been a recent thrust, however, towards developing intrinsically interpretable (ante-hoc interpretable) models that render models to be interpretable in the training process itself (Rudin, 2019; Vilone & Longo, 2021; Nauta et al., 2023). These recent efforts have largely focused on traditional supervised learning; methods that can learn continually and are also inherently interpretable remain largely unexplored.

A class of interpretable models that has gained prominence in recent years is *concept-based learning*. A concept is typically a high-level, inherently interpretable unit of information. A class can be abstracted into a set of concepts that define the characteristics of that class. For example, the class *cat* may be broken down into the concept set {*fur, whiskers, four legs, pointy ears, sociable*}. Such concept-based approaches allow atomic concepts to be combined to signify the presence of a particular class in image-based tasks. Recent works such as (Koh et al., 2020; Oikarinen et al., 2023; Yang et al., 2023) have shown promise in using concept-based models to enhance interpretability of image classification models, but however have not been studied for learning continually. When using concept-based models in a continual learning setting, several new problems emerge: (i) from

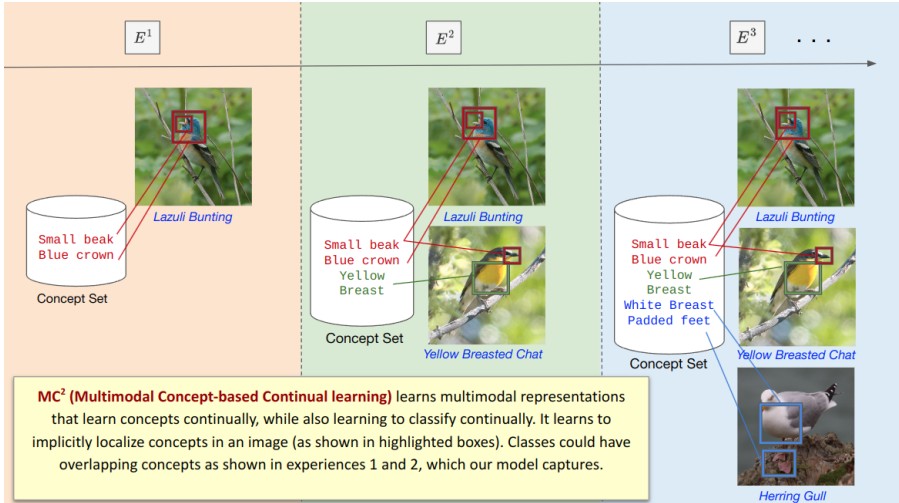

Figure 1: Illustration of our model and the proposed setting

a concept-based learning perspective, the model has to allow mechanisms to learn new concepts without forgetting past concepts; (ii) from a continual learning perspective, in addition to learning new classes over time (standard continual learning), the model also has to learn new concepts over time, i.e. the model has to address catastrophic forgetting in classes as well as concepts; and (iii) it is possible that older concepts may be component of a newer class in a later task; it has to learn these associations effectively too. These challenges make this setting a non-trivial one, and timely considering the focus on ante hoc interpretable models. A very recent work (Rymarczyk et al., 2023) addressed this setting for the first time, also supporting the need for this direction of work. However, the notion of a concept is different from earlier efforts in this work, and is oriented towards part-based prototypes. Such part-based structures may not capture abstract concepts or relationships such as, for e.g., *cat* and *sociable*. We focus on a more generic approach to concepts that are not necessarily part-based but text-based high-level semantics associated with an object category.

To this end, we propose $\mathbf{MC^2}$, a novel multimodal concept-based continual learner that not only accommodates new classes and concepts, but also implicitly localizes text-based concepts in images. We consider text encodings of text concepts, which we call concept anchors, along with image representations to create a set of *multimodal concepts* for a given image. These multimodal concepts are latent vectors which contain information that help in classification while also providing interpretations. We introduce the notion of *concept grounding*, which allows the interpretation of multimodal concepts in terms of text-based concepts. We also design $\mathbf{MC^2}$ with the consideration that it should be able to learn continually. Our proposed approach is not limited to a pre-specified number of concepts and classes, thus making it scalable by design for class-incremental and concept-incremental learning. Our key contributions are summarized below:

- We propose a novel method for concept-based continual learning that can adapt continually to new classes as well as new concepts, without increasing the number of parameters. Standard experience replay does not help reliably explain the model in terms of concepts; we hence introduce a new concept-augmented exemplar replay approach that allows the model to retain concept-based explanations of previous experiences.

- We propose *multimodal concepts*, a combination of image embeddings and interpretable concept anchors, to perform classification. These multimodal concepts are grounded to their corresponding text-based concept anchors, thus making them interpretable. We also show that the vision-language models used in our approach need not be pre-aligned, allowing for more flexibility in the method.

- Our approach offers *multi-hoc* concept-based interpretability, i.e. it is designed in an ante-hoc fashion to offer interpretability in the form of high-level concepts, and can also be employed as a concept-specific attribution method, which enhances post-hoc interpretability by identifying regions of interest involved in the search for a specified concept. For example, in Figure 3, we show that our model is able to reliably localize image-level attributions for queried concepts.

- We perform a comprehensive set of experiments to evaluate our proposed method on well-known benchmark datasets, and also compare our method against continual adaptations of earlier concept-based methods. We study our method's performance both in a continual as well as full-data setting. We perform qualitative evaluations of how well our model learns to associate concepts with localized visual cues in images, and also study the goodness of concepts by demonstrating their effectiveness in post-hoc interventions.

## 2 RELATED WORK

**Interpretability of Deep Neural Network Models**: Interpretability methods in DNN models can be broadly classified into post-hoc and ante-hoc methods. Post-hoc methods aim to interpret model predictions through several strategies, including *Gradient-weighted Class Activation Mapping-based* methods which focus on highlighting influential features by tracking gradient flows to the final layer (Selvaraju et al., 2017; Chen et al., 2020a; Chattopadhay et al., 2018); *Integrated gradient- based* methods that compute the gradient integration via the Riemann integral (Sattarzadeh et al., 2021; Yvinec et al., 2022; Benitez et al., 2023); *Shapley value-based* methods that address model interpretation using Shapley values (Sundararajan & Najmi, 2020; Wang et al., 2020a; Jethani et al., 2021; Wang et al., 2020a), and several other *Non-gradient based* methods(Dabkowski & Gal, 2017; Fong & Vedaldi, 2017; Petsiuk et al., 2018; Montavon et al., 2019). While post-hoc methods offer insight into the model's interpretability without posing additional model constraints, recent efforts have highlighted the issues with post-hoc methods and their reliability in reflecting a model's reasoning (Rudin, 2019; Vilone & Longo, 2021; Nauta et al., 2023). Besides, When interpretations are inaccurate, it becomes difficult to reason whether the problem lies with the interpretation method or if the model relied on spurious correlations in the data. There have also been concerns on post-hoc interpretability and its larger success only on simple model architectures (Burns & Steinhardt, 2021; Adebayo et al., 2021; Bordt et al., 2022). On the other hand, ante-hoc methods that jointly learn to explain and predict provide models that are inherently interpretable (Sokol & Flach, 2021; Benitez et al., 2023). Ante-hoc methods have also been found to provide interpretatations that help make the model more robust and reliable (Alvarez-Melis & Jaakkola, 2018; Chattopadhyay et al., 2022). We focus on this genre of methods in this work.

**Continual Learning**: Continual learning (CL) methods aim to tackle catastrophic forgetting (Hadsell et al., 2020) using techniques that alleviate forgetting across experiences. These methods have been extensively studied in the last few years and can be broadly grouped into three main categories: *exemplar replay-based* methods use a small exemplar buffer to store highly-representative samples of classes belonging to previous experiences using some similarity metric (Shin et al., 2017; Mi et al., 2020; Van de Ven et al., 2020; Maracani et al., 2021; Graffieti et al., 2023). Variations of such methods tend to adapt gradient-based sample selection strategies for populating the buffer (Aljundi et al., 2019; Jin et al., 2020; Tiwari et al., 2022). *Architecture-based* methods on the other hand rely on strategies such as network expansion and require updating parameters of the model as new classes arrive (Ebrahimi et al., 2020; Douillard et al., 2022; Kang et al., 2023), such methods can be costly and difficult to scale. *Regularization-based* methods tend to protect influential weights from old experiences from mutation (Sha et al., 2016; Jung et al., 2020; Maschler et al., 2021; Li et al., 2023). However, methods for interpretable continual learning have largely remained unexplored, except for one very recent work (Rymarczyk et al., 2023), discussed later in this section.

**Concept-based Interpretability**: Koh et al. (2020) proposed Concept Bottleneck Models (CBMs), a method that uses interpretable, human-defined concepts, combining them linearly to perform classification. CBMs also allow human interventions on concept activations (Shin et al., 2023; Steinmann et al., 2023) to steer the final prediction of the model. Subsequent efforts such as (Marconato et al., 2022b; Havasi et al., 2022; Barker et al., 2023) improved upon specific issues such as concept leakage. Adaptation of concept-based learning to provide ante-hoc interpretability to any DNN architecture was also shown in (Sarkar et al., 2022). While the presence of a representative set of concepts helps with interpretability, collecting such dense concept annotations is time-consuming. This issue was addressed in (Kim et al., 2023; Collins et al., 2023; Yan et al., 2023) where the intermediate semantic concepts are obtained by replacing domain experts with Large Language Models (LLMs). This allows for ease and flexibility in obtaining the concept set, while also overcoming the issue of concept leakage using concept filters. We follow this approach to obtain concepts in this work too. Besides making concept-based learning more feasible, using LLMs to obtain concepts also allow grounding of neurons in a bottleneck layer to a human-understandable concept, an

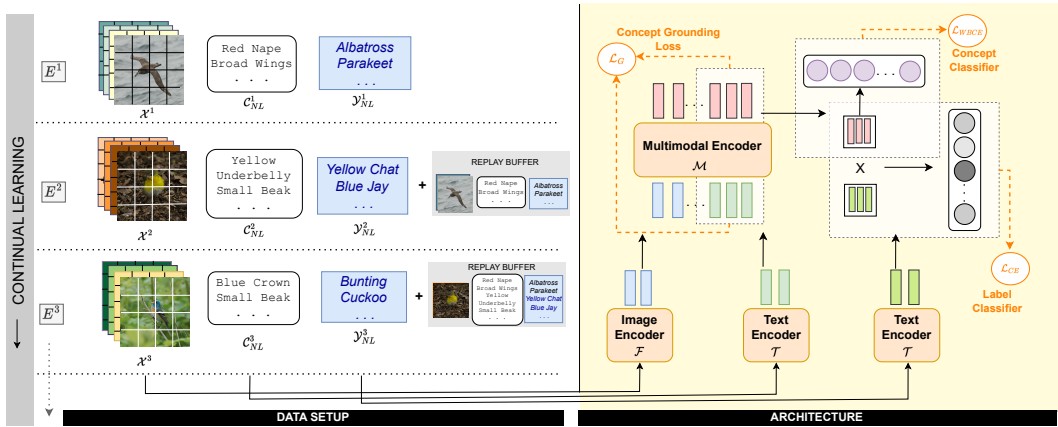

Figure 2: **Overview of our data setup and proposed architecture.** Our architecture receives new classes and associated concepts across multiple experiences in a continual learning setting. We use pre-trained language and vision encoders to get embeddings for the input image, concepts, and classes. These are then used to create multimodal concepts using our *Multimodal Encoder*. These multimodal concepts are grounded to their anchor concepts using a loss function, and are used to predict both the class label and the presence/absence of corresponding concepts in the image.

issue with CBMs that was highlighted in (Margeloiu et al., 2021). Other concept-based methods (Alvarez-Melis & Jaakkola, 2018; Chen et al., 2020b; Kazhdan et al., 2020; Rigotti et al., 2021; Benitez et al., 2023) use a different notion of concepts based on prototype representations of image features; we follow the former approach in this work. Importantly, all aforementioned efforts only perform concept-based learning in the traditional supervised setting, with no explicit efforts towards addressing the continual learning setting.

**Interpretable Continual Learning**: As stated earlier, we focus on the premise that making models continual and interpretable allows them to adapt their reasoning mechanisms to unseen data that arrive over time. Existing concept-based models (Koh et al., 2020; Oikarinen et al., 2023; Yang et al., 2023) address interpretability under the assumption that classes and concepts are pre-defined, making the concept set rigid. Concept-based continual learning has remained largely unstudied. We identify (Marconato et al., 2022a) as an early effort in this direction; however, this work trains CBMs in a continual setting under an assumption that all concepts, including those required for unseen classes, are accessible from the first experience itself, which does not emulate a real-world setting. More recently, Rymarczyk et al. (2023) proposed a method that is both continual and interpretable that uses part-based prototypes as concepts. As mentioned earlier, our notion of concepts allows us to go beyond parts of an object category, as in CBM-based models.

## 3 MC²: METHODOLOGY

**Preliminaries and Notations.** Given a sequence of experiences $\{E^1, E^2, ..., E^T\}$, with each experience $E^i$ consisting of $n$ image-label pairs $(\mathcal{X}^i, \mathcal{Y}^i) = \{(x^i_1, y^i_1), (x^i_2, y^i_2), ..., (x^i_n, y^i_n)\}$, a class-incremental continual learning (CiCL) system aims to learn a task $E^t$ without catastrophically forgetting tasks $E^1$ to $E^{t-1}$. In the scenario where human-provided concepts are used for classification, each experience $E^i$ consists of $n$ image-label-concept tuples $(\mathcal{X}^i, \mathcal{Y}^i, \mathcal{C}^i) = \{(x^i_1, y^i_1, \mathcal{C}^i_1), (x^i_2, y^i_2, \mathcal{C}^i_2), ..., (x^i_n, y^i_n, \mathcal{C}^i_n)\}$, where $\mathcal{C}^i$ is the set of concepts known during experience $E^i$ and $\mathcal{C}^i_k$ is the set of active concepts in example $k$. The set of concepts known during $E^i$ is the union of all concept sets from task $E^1$ to $E^i$. For the following sections, we use the subscript *NL* for an object if it is presented to our method in natural language.

*Concept Annotations:* The natural language concepts in $\mathcal{C}^i$ may be provided as part of the dataset (e.g. CUB dataset). However, collecting concept annotations for classes can be tedious in general, especially if the number of classes is very large or if suitable and sufficient domain experts are not available. In such cases, one can derive the concepts by querying a Large Language Model (LLM) as proposed by Oikarinen et al. (2023) and Yang et al. (2023). Our approach is inclusive of both these approaches, depending on what may be available for a given dataset.

Our model learns to create multimodal concept embeddings that are grounded to their corresponding textual anchors and also contain the corresponding visual information that together aid in classification. We now formally define the terms *Grounding* and *Anchor*, as used in our work.

**Definition 1.** *Given a vocabulary $\mathcal{V}$ containing words, phrases, or sentences in natural language, a text encoder $\phi : \mathcal{V} \to \mathbb{R}^d$, a vector $\mathbf{u} \in \mathbb{R}^d$, and a distance function $\mathcal{D} : \mathbb{R}^d \times \mathbb{R}^d \to \mathbb{R}$, $\mathbf{u}$ is* **grounded** *to a word, phrase, or sentence $v_{NL} \in \mathcal{V}$ if $\mathcal{D}(\mathbf{u}, \phi(v_{NL})) \leq \varepsilon$ for some distance function $\mathcal{D}$ and $\varepsilon > 0$. Then, $v_{NL}$ is said to be an* **anchor** *of $\mathbf{u}$.*

In other words, a feature vector is said to be *grounded* to a text term if their embeddings align with a certain tolerance $\varepsilon$. We present an overall schematic of $\mathbf{MC^2}$ in Figure 2. Our approach leverages embeddings obtained from both the input image and the user-defined concepts present in that image. These are used to create multimodal concept embeddings using textual concept embeddings as anchors. Our algorithm comprises three major components: a *multimodal concept encoder*, a *concept-grounding module*, and a *concept-augmented experience replay*, each of which is described in detail below.

**Multimodal Concept Encoder.** Our proposed setting requires each sample of experience $E^t$ to be of the form $(x_i \in \mathcal{X}^t, y_i \in \mathcal{Y}_{NL}^t, \mathcal{C}_i \in \mathcal{C}_{NL}^t)$, where $x_i$ is an input image and $\mathcal{C}_i$ is the concept set for the current experience in natural language. $y_i$ is the corresponding class name, also in natural language. The embeddings for the classes and concepts in $\mathcal{Y}_{NL}^{(i)}$ and $\mathcal{C}_{NL}^{(i)}$ are obtained using a pre-trained language encoder. Formally, given an image $x_i$ and a feature extractor $\mathcal{F}$, the image embedding $\mathbf{x}_i$ is obtained as $\mathbf{x}_i = \mathcal{F}(x_i)$; similarly, given a concept word or phrase $c_j \in \mathcal{C}_{NL}^{(i)}$ and a text encoder $\mathcal{T}$, the text embedding $\mathbf{c}_j$ is obtained as $\mathbf{c}_j = \mathcal{T}(c_j)$.

In order to enable cross-modal learning, we create multimodal representations of the image and text inputs using their respective embeddings. This allows the learned representation to exchange information between the modalities, and also assimilate information about the occurrence of the textual concept in the provided image. To this end, we use a multimodal encoder $\mathcal{M}$, which is a stack of transformer encoder layers. We provide the image embeddings $\mathbf{x}_i$ as well as the concept embeddings $\mathbf{c}_1 \mathbf{c}_2 ... \mathbf{c}_{|\mathcal{C}_{NL}^i|}$ as an input sequence to $\mathcal{M}$. The output of $\mathcal{M}$ is also a sequence of vectors, wherein we map the last $|\mathcal{C}_{NL}^i|$ of the sequence to the concept anchors using the concept grounding module (described later in this section). A shared sigmoid-activated linear layer, $\sigma(\cdot)$, is also trained on each multimodal concept vector to perform binary classification, where 1 indicates the presence of a concept, and 0 otherwise. A weighted binary cross-entropy loss, $\mathcal{L}_{WBCE}$, is used to train the model for concept classification. We also classify the entire image represented by these multimodal concepts using the standard image-level cross-entropy loss, $\mathcal{L}_{CE}$. The loss for training $\mathcal{M}$ is then a weighted sum of these two losses: $\mathcal{L} = \mathcal{L}_{CE} + \lambda \mathcal{L}_{WBCE}$, where $\lambda$ is a hyperparameter. Empirically, we find that $\lambda = 5$ works marginally better than lower or higher values. More details on $\mathcal{L}_{WBCE}$ are provided in the appendix.

*Classification using the Multimodal Concept Encoder:* The alignment between the $j$th multimodal concept vector of the current sample, $\mathbf{c}_j'$, and the embedding of the $k$th class $\mathbf{y}_k$ of the current experience can be obtained by taking the dot product of the two vectors. We define the strength $s_k$ of class $k$ in a given image to be the sum of dot products of all multimodal concepts onto the class embedding of $k$, i.e. $s_k = \sum_{j=1}^{|\mathcal{C}^i|} \mathbf{c}_j' \cdot \mathbf{y}_k$. The classification result is then given by: $\text{argmax}_y(s_1, s_2, ..., s_{|\mathcal{Y}^i|})$, that is, the index of the class having the greatest strength with respect to all concepts. We use $s_k$ as the logit of class $k$, and perform a softmax operation on top of the logits to get classification probabilities, which are used to train the model with the standard cross-entropy loss $\mathcal{L}_{CE}$. It should be noted that deriving class strengths from the multimodal concepts does not require any additional parameters; this enables scalability of our approach to unseen classes and concepts when deployed in a continual setting.

**Concept-Grounding Module.** While the outputs of $\mathcal{M}$ are multimodal by design, they do not implicitly provide explanations in terms of human-defined concepts. The concept-grounding module allows us to ground these multimodal concept vectors to known concept anchors that are directly obtained from textual descriptions of concepts. We use the last $|\mathcal{C}_{NL}^i|$ vectors of the output sequence given by $\mathcal{M}$ as the set of our multimodal concepts. This allows us to create a one-to-one mapping between input and output concept vectors. We use a *Concept Grounding Loss*, $\mathcal{L}_G$, to ground the

predicted multimodal concepts with their corresponding concept anchors, as below:

$$\mathcal{L}_G = -\frac{1}{|\mathcal{C}^i|} \sum_{k=1}^{|\mathcal{C}^i|} cos(\mathbf{c}_k, \mathbf{W}^T \mathbf{c}'_\mathbf{k} + \mathbf{b}) = -\frac{1}{|\mathcal{C}^i|} \sum_{k=1}^{|\mathcal{C}^i|} \frac{\mathbf{c}_k . (\mathbf{W}^T \mathbf{c}'_k + \mathbf{b})}{|\mathbf{c}_k| . |\mathbf{W}^T \mathbf{c}'_k + \mathbf{b}|} \tag{1}$$

where $\mathbf{c}'_k$ represents the multimodal vector corresponding to concept anchor $\mathbf{c}_k$. $\mathbf{W}$ and $\mathbf{b}$ are learnable parameters that are shared among all concepts and serve to perform concept alignment. Grounding the multimodal concept vectors enables them to encode the association between the corresponding concept anchor and the given input image.

**Concept-Augmented Experience Replay.** Experience (or exemplar) replay is a standard technique used in continual learning to prevent catastrophic forgetting. This is typically implemented by creating a (small) memory buffer that contains training samples from past experiences. When a model is trained on a new experience, the memory buffer is sampled and the model is additionally trained on these stored samples. We propose an extension called Concept-Augmented Experience Replay in this work, wherein we store the class-level concept labels in addition to images and class labels. The exemplar loss is identical to the loss $\mathcal{L}$ used to train $\mathcal{M}$, i.e. the concept-level loss in the multimodal concept encoder $\mathcal{L}_{WBCE}$ is additionally used on these concepts when replaying these buffer samples, in addition to the cross-entropy loss. While this simple enhancement of experience replay may seem trivial since one can simply ignore concepts in the buffer, we show later in the paper that the quality of concepts learned through concept-augmented experience replay is far superior to standard experience replay that does not store concepts (see Table 4).

## 4 EXPERIMENTS AND RESULTS

We perform a comprehensive suite of experiments to study the performance of $\mathbf{MC^2}$ on well-known benchmarks that allow us to study both continual as well as concept-based learning: CIFAR-100, ImageNet-100, and CalTech-UCSD Birds 200 (CUB200). We study our method both in a continual setting as well as in a full-data setting. We also examine different components of our model and study each of their importances to the method. Details related to architecture implementation and hyperparameter selection have been described in the appendix.

**Performance Metrics.** In the class-incremental setting, we follow earlier literature in using two metrics to evaluate the performance of different methods. *Final Average Accuracy (FAA)* is a measure of how well a model has adapted to a sequence of tasks or data streams over time. It represents the average accuracy of the model on the validation splits of all tasks or data streams after it has completed its learning process, across the experiences. FAA is defined as: $FAA = \frac{1}{T} \sum_{i=1}^{T} acc_i^T$, where $acc_i^T$ represents the model's accuracy on the validation split of experience $i$ after training on $T$ experiences. *Average Forgetting (AF)* quantifies the extent to which a model forgets previously learned knowledge when exposed to new experiences. It measures the drop-in performance on tasks learned in previous experiences after the model has been trained on newer experiences. Lower average forgetting indicates better model stability and performance in a continual learning scenario. AF at task $T$ is defined as: $AF = \frac{1}{T-1} \sum_{i=1}^{T-1} acc_i^i - acc_i^T$, i.e., the difference in accuracy on the validation set of task $i$ when it was originally learned and the accuracy on it after the model has been trained on $T$ experiences. In the full-data setting where the model is provided with all training data in a single experience, we use the standard *Classification Accuracy* to evaluate different methods, viz. the ratio of correctly classified examples and the total number of examples.

**Baselines.** While there has been very little effort on explicitly studying concept-based continual learning, we thoroughly evaluate our approach in class-incremental and full-data settings by comparing it with existing works that use human-defined concepts. Our baseline methods for comparison include: (i) (Marconato et al., 2022a), which uses a concept bottleneck layer with one neuron assigned to each concept. In a class-incremental setting this baseline makes the assumption that all concepts, including those that would ideally only be provided in future experiences, are provided upfront. The model then uses a growing linear layer with new neurons added for new classes for the final classification; (ii) *Incremental CBM*, a version of the Concept Bottleneck Model (Koh et al., 2020) that we modify to adapt to a class-incremental and concept-incremental learning scenario. We grow both the bottleneck layer and the linear classification layer as new classes and new concepts are introduced. One can see that this is a generalized version of the previous baseline where the assumption that all concepts are provided upfront is relaxed; We also consider (iii) and (iv) which are

*Label-Free CBM* (Oikarinen et al., 2023) and *LaBo* (Yang et al., 2023), variations of CBM that use embeddings of natural language concepts as the bottleneck layer. Both these methods also propose ways to discover concepts for a specified class by querying LLMs. They primarily differ between them in how they query the LLM and filter the obtained concept set. We adapt these methods to the continual learning setting by allowing concepts from previous experiences to be considered when a new experience is provided.

**Implementation Details.** For ImageNet-100 and CIFAR-100, we grow the concept set at every experience as new concepts arrive, while discarding duplicate concepts from the new set. We show the number of concepts, with and without duplicates, in Table 1. In the case of CUB, the number of concepts is fixed to 312 across all experiences (as provided with the dataset). The number of concepts can be reasonably large, as shown. This can cause out-of-memory errors when used with the standard attention mechanism since our method performs attention over the entire concept set. To address this, we also study a simple variant of our method with linear attention, which we denote as $MC^2$(Linear) in our results. More details about the use of linear attention are provided as part of our ablation studies. Other implementation details including dataset details, hyperparameters, and training setups are provided in the Appendix. Our code will be made publicly available on acceptance.

| Exp | CIFAR-100 | ImageNet-100 |
|-----|-----------|--------------|
| E1  | 257 (257) | 214 (214) |
| E2  | 460 (527) | 359 (416) |
| E3  | 638 (794) | 457 (594) |
| E4  | 798 (1046) | 545 (762) |
| E5  | 925 (1309) | 641 (945) |

Table 1: Number of concepts per class, excluding duplicates across experiences (Exp) (inclusive numbers in parentheses)

| Method | CIFAR-100 | | CUB | | ImageNet-100 | |
|--------|-----------|-----|-----|-----|--------------|-----|
| | FAA | AF | FAA | AF | FAA | AF |
| **CBM** (Koh et al., 2020) | 0.4333 | 0.5646 | 0.5875 | 0.2029 | 0.4523 | 0.5553 |
| **CBM (Sequential)** (Koh et al., 2020) | 0.3533 | 0.6025 | 0.5329 | 0.1347 | 0.4523 | 0.5553 |
| ICIAP (Marconato et al., 2022a) | 0.4196 | 0.5719 | 0.5875 | 0.2029 | 0.4689 | 0.5253 |
| **ICIAP (Sequential)** (Marconato et al., 2022a) | 0.2945 | 0.5937 | 0.5329 | 0.1347 | 0.4689 | 0.5253 |
| **Label-Free** (Oikarinen et al., 2023) | 0.3200 | **0.2338** | 0.1934 | 0.4408 | 0.1493 | 0.2760 |
| **LaBo** (Yang et al., 2023) | 0.3009 | 0.6879 | 0.3101 | 0.4741 | 0.3384 | 0.4560 |
| $MC^2$ | **0.7022** | 0.3003 | 0.8137 | 0.0611 | 0.7970 | 0.0877 |
| $MC^2$**(Linear)** | 0.6920 | 0.3142 | **0.8188** | **0.0531** | **0.7985** | **0.0776** |

Table 2: Continual learning performance of different methods over 5 experiences

**Quantitative Results.** Table 2 shows our results on concept-based continual learning. On CIFAR-100 and ImageNet-100, our approach outperforms all baselines by a significant margin. It should be noted that this is done *without adding any additional parameters* to our model with newer experiences, whereas other methods require new parameters to incorporate new classes and concepts. We also observed significantly lower forgetting across experiences using our approach. These results show that our model can readily incorporate knowledge about new concepts and classes while internally forming the required concept-class associations. It is also able to remember these associations to a good extent, even after being trained on new tasks.

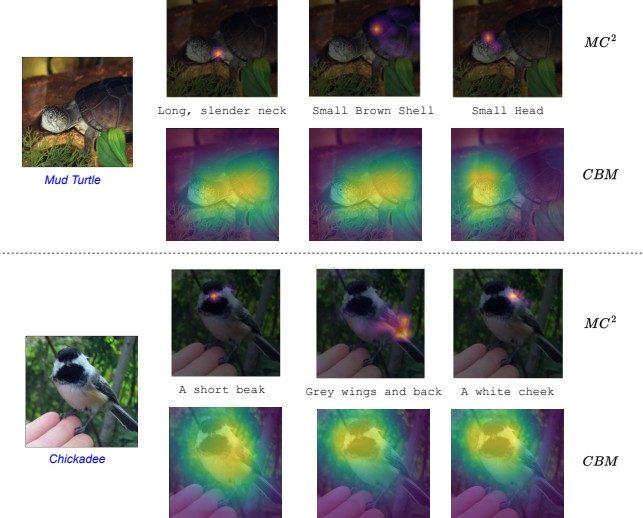

Figure 3: **Visual grounding of concepts:** Qualitative results for localizing concepts using $MC^2$ versus when localizing the same concepts using GradCAM on CBMs

**Qualitative Results.** *Visual Grounding and Attributions.* We extend our method as a post-hoc analysis tool to provide visualizations of the attention maps learned by our model. As shown in Figure 3, each heatmap shows the region an attention head focuses on for a specified concept. It can be seen that our model learns to assign a subset of its attention heads to extract user-defined concepts from an image. This is in contrast with models that do not provide such grounding, which fail to reliably extract user-defined concepts from the given image (Margeloiu et al., 2021).

**More Results: Full-Data Training.** To see how our model fares in standard classification settings, we evaluate our method in a full-data, single-experience setting on three datasets. These results for presented in Table 3. In this setting, we find that $MC^2$ considerably outperforms the next closest baseline on the CUB dataset, indicating that it is highly effective when used to differentiate between fine-grained classes. It also achieves comparable performance on ImageNet-100 and CIFAR-100, even though this setting is not our focus.

| Method | CIFAR-100 | CUB | ImageNet-100 |
|---|---|---|---|
| CBM-J | 0.7868 | 0.7231 | 0.7773 |
| CBM-Seq | 0.5712 | 0.6932 | 0.4265 |
| Label-Free | 0.6431 | 0.7413 | 0.7818 |
| LaBo | **0.8572** | 0.7015 | **0.8506** |
| Ours | 0.8567 | **0.8401** | 0.8466 |

Table 3: Classification performance of different methods in the full-data (single experience) setting. CBM-J involves joint training of the CBM as in Koh et al. (2020), while CBM-Seq involves sequential training.

**More Results: Evaluating Concepts.** The concepts given by LLMs (in the case of ImageNet-100 and CIFAR-100), as well as concepts annotated by humans (as in CUB200), can be noisy. Therefore, directly comparing accuracies of concept classification may not evaluate how well the networks learn concepts. We instead evaluate the learned concepts in two ways: (i) using concept neurons (inspired by Marconato et al. (2022a)), and (ii) using interventions, each of which is described below.

*Evaluating goodness of concepts through concept neurons*: A concept neuron (see Marconato et al. (2022a)) predicts the presence or absence of a given concept based on a grounded concept representation. After training, such concept neurons should be able to feed a linear classifier on par with the grounded concept vectors. We evaluate this

| Dataset | FAA w/ CR | Linear Acc w/ CR | FAA w/o CR | Linear Acc w/o CR |
|---|---|---|---|---|
| CIFAR-100 | 0.7022 | 0.7650 | 0.6722 | 0.4511 |
| CUB200 | 0.8137 | 0.7914 | 0.7844 | 0.1382 |
| ImageNet-100 | 0.7970 | 0.7722 | 0.7903 | 0.5903 |

Table 4: Linear layer training on top of concept neurons; CR = concept-augmented experience replay

by treating a group of concept neurons as a bottleneck layer and training a linear classifier on top of the neurons. Since this only examines concepts, we train the linear layer on all classes simultaneously for 3 epochs, irrespective of whether the model was trained incrementally or in a full data setting. The results are shown in Table 4 with and without concept-augmented experience replay (CR). Evidently, the concepts perform significantly better when using our proposed CR replay.

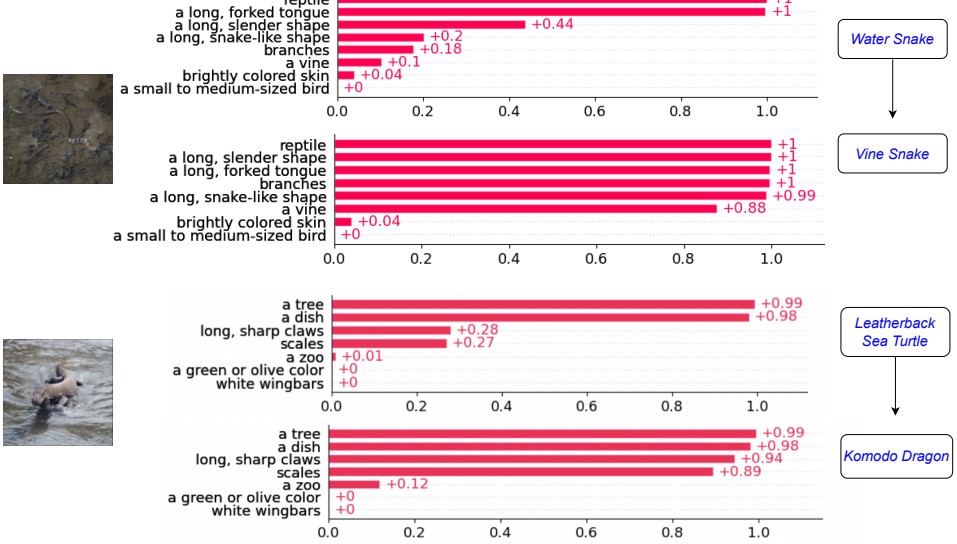

Figure 4: **Manual interventions on concepts:** We identify concepts that are incorrectly labeled, and modify them based on the image semantics, this results in correct classification.

*Evaluating concepts using interventions*: Interventions allow us to study the (potentially causal) relationship between concepts and the classes they describe. To study these, we use the linear layer trained above to evaluate how well our model learns such concept-class relationships. We consider samples that are misclassified by the newly trained linear layer and perform interventions on the wrongly predicted concepts using the mechanism described in (Koh et al., 2020). Figure 4 shows qualitative results of performing interventions on a misclassified image. We observe that most concepts present in an image are usually correctly identified, but performing interventions on a few key misclassified concepts results in correct classifications a majority of the time. This highlights the goodness of semantics of the learned concepts, and its impact on classification. In the figure, we see that the image of *Komodo Dragon* in also activates the concept "*a tree*" due to visual cue similarity. When the key concepts "*scales*" and "*long, sharp claws*" are activated better, the model now classifies this correctly as a *Komodo Dragon*.

**Ablation Studies: Vision-Language Alignment.** We now study the importance of having pre-aligned vision and text encoders to get image, class, and concept embeddings. Alignment here refers to the property that for a given image and corresponding image description in natural language, the encoders produce vectors that are *close* in a high-dimensional space based on some predefined metric. We perform a grid search over 9 different vision-language encoder pairs. Two of these pairs, CLIP (Radford et al., 2021) and FLAVA (Singh et al., 2022), have pre-aligned vision-language encoders. We also used BERT (Devlin et al., 2018) and ViT (Dosovitskiy et al., 2021) models trained on unimodal data, where our model explicitly aligns the modalities.

The results are shown for CUB in Table 6, and for ImageNet-100 in Table 5. Our results indicate that using pre-aligned vision-language (VL) models for our method; our method's alignment for this task is in fact superior to pre-aligned models. This is because pre-aligned VL models are trained at a general image level, while our explicit approach allows more fine-grained association between image and text.

| Text
Vision | FLAVA | CLIP | BERT |
|---|---|---|---|
| FLAVA | 0.7036 | 0.6532 | 0.6952 |
| CLIP | 0.7372 | 0.7247 | 0.7125 |
| ViT | **0.7970** | **0.7458** | **0.7404** |

Table 5: VL alignment, ImageNet100

**Ablation Studies: Attention Mechanism and Scalability.** In its naive implementation, the compute requirements of our method can grow quadratically with the number of concepts. This is due to the quadratic dependency of the vanilla attention mechanism used in transformer blocks. Fortunately, recent attempts (Katharopoulos et al., 2020; Vyas et al., 2020; Shen et al., 2021; Wang et al., 2020b; Kitaev et al., 2019) have been made to improve the computational efficiency of transformer

| Text
Vision | FLAVA | CLIP | BERT |
|---|---|---|---|
| FLAVA | 0.7628 | 0.6501 | 0.7218 |
| CLIP | 0.8047 | 0.7180 | 0.7970 |
| ViT | **0.8245** | **0.7973** | **0.8344** |

Table 6: VL alignment, CUB

architectures. As stated earlier, we propose a viable variant to make our model practically feasible for a large number of concepts: $MC^2$ with Linear Attention, whose compute requirements grow linearly with the number of concepts. We use transformer blocks featuring the linear attention mechanism proposed in (Katharopoulos et al., 2020) as a drop-in replacement in our multimodal encoder. These results are also shown in Table 2. We see that using linear attention surprisingly achieves better results on CIFAR-100 while achieving comparable performance to vanilla attention on CUB and ImageNet-100.

# 5 CONCLUSIONS AND FUTURE WORK

In this work, we propose a new perspective to integrating human-defined concept-based models perform in a continual setting. We propose a method that uses pre-trained language and vision encoders to create multimodel concepts, which are anchored to natural language concepts. Our approach can reliably interpret classification results in terms of the provided concepts, and can incorporate new concepts and classes at a later time as well. We perform comprehensive evaluations of our method on three benchmark datasets and also study the efficacy of concepts in our pipeline. Our qualitative and quantitative results show the usefulness of the proposed method. Although our method provides a high-performing continual and interpretable model, the use of a pre-trained vision encoder limits us from using arbitrary augmentations (e.g. color jitter) to improve model generalization. Allowing for this in unaligned unimodal encoders could help further performance. From an interpretability viewpoint, developing an improved intervention mechanism that can be used on our model without an explicit linear layer would be an interesting direction of future work. We can also explore other forms of attention, such as Flash Attention (Dao et al., 2022), to improve the practical scalability of our method.

**Reproduciblity Statement:** Necessary details required to reproduce our results have been provided in the Appendix. The full code shall be released publicly upon acceptance of the paper.

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

# A  APPENDIX

## CONTENTS

### A1.1  ARCHITECTURE AND IMPLEMENTATION DETAILS

*Information about Vision, Language, and Multimodal Encoders ($\mathcal{F}, \mathcal{T}, \mathcal{M}$).* We use the FLAVA (Singh et al., 2022) language encoder paired with the ViT (Dosovitskiy et al., 2021) image encoder. Each encoder has a latent embedding dimension of 768. Our multimodal encoder uses two stacked transformer blocks with the same latent embedding size. We use the *HuggingFace*[1] library to implement the transformer in case of the Full Attention version, or the *fast-transformers*[2] library to implement transformer blocks with Linear Attention. The experimental setup is implemented using *PyTorch*[3].

*Training hyperparameters.* For Cifar100, we train the model for 10 epochs in every experience, with a starting learning rate of 0.001 and a batch size of 48. In the case of CUB, we train our model for 25 epochs in every experience and stop training after 15 epochs if the model converges. We start with a learning rate of 0.0003 and a batch size of 64. In the case of Imagenet100, we train the model for 5 epochs in every experience, with a starting learning rate of 0.001 and batch size of 48. In all three cases, we use Cosine Annealing to schedule the learning rate, decaying it down to 0.0001.

*Details of $\mathcal{L}_{WBCE}$.* For a given image, the ratio of the number of active concepts to the number of total concepts is quite small. This necessitates penalizing the misclassification of active concepts more strongly than the misclassification of inactive concepts. We do this by weighting the loss for active concepts by the fraction of *inactive concepts*, and weighting the loss for inactive concepts by the fraction of *active concepts*. The loss $\mathcal{L}_{WBCE}$ is then defined as:

$$\mathcal{L}_{WBCE} = \frac{\text{\# inactive concepts}}{\text{\# concepts}} \sum_{i=1}^{|\mathcal{C}^{\text{active}}|} \mathcal{L}_{BCE}(\sigma(\mathbf{c}'_i), 1)$$
$$+ \frac{\text{\# active concepts}}{\text{\# concepts}} \sum_{i=j}^{|\mathcal{C}^{\text{inactive}}|} \mathcal{L}_{BCE}(\sigma(\mathbf{c}'_j), 0)$$

*Additional Details about Concept Bottlenecks.* A Concept Bottleneck layer, introduced by **?**, is a layer where each neuron corresponds to a specific concept. Models containing such bottlenecks can be trained sequentially or jointly with the classification layer. Sequential and joint settings are applicable when the model contains a bottleneck layer followed by a classification layer. In the sequential setting, the model is first trained to predict concept labels. Post-training, a classifier is trained on top of concept logits predicted for the input images. The model and classifier are optimized separately. In joint training, both concept predictions and the classifier are trained end-to-end and optimized jointly.

### A1.2  DATASET DETAILS

*Descriptions of Datasets.* **Cifar100** consists of 50000 training images and 10000 validation images spanning 100 classes. Each image is a 3-channel RGB image of size 32x32 pixels. Concept annotations per class are not provided, and hence we query a Large Language Model as described by

---

[1]https://huggingface.co/

[2]https://fast-transformers.github.io/

[3]https://pytorch.org/

(Oikarinen et al., 2023) to get the concept set, excluding the concept filters applied post-training for reduction in the number of concepts. We get a total of 925 concepts for Cifar100. **CUB200 (CalTech-UCSD Birds 200)** is a fine-grained bird identification dataset consisting of 11000 RGB images of 200 different bird species. In this case, the concept annotations are provided by human annotators. All concepts are shared among a few classes, which means that the entire concept set is available from the first experience itself. This gives us a platform to show that our method gives state-of-the-art results even on fine-grained visual classification in a much simpler setting where existing methods still fail to perform comparably. **ImageNet100** is a subset of the Imagenet1K dataset consisting of 100 classes with 1300 training images and 50 validation images per class. The subset includes both coarse and fine-grained classes. We choose classes such that each class in a new experience adds new concepts, in addition to using concepts available from past experiences. We provide a subset of some classes and concepts per dataset in A10. To use the datasets in a continual setting, we split each dataset into 5 tasks having overlapping concept sets. Details of the number of concepts in each experience for Imagenet100 and Cifar100 have been provided in table 1 of the main paper.

### A1.3 PER-EXPERIENCE PERFORMANCE

Here, we report the performance of $MC^2$ and baseline methods across five experiences, providing the values of Average Accuracies and Forgetting at every experience.

| Model | Experience 1 | | Experience 2 | | Experience 3 | | Experience 4 | | Experience 5 | |
|---|---|---|---|---|---|---|---|---|---|---|
| | AA | Forget | AA | Forget | AA | Forget | AA | Forget | AA | Forget |
| CBM-Seq | 0.8744 | 0.6607 | 0.6764 | 0.6338 | 0.5292 | 0.5793 | 0.4392 | 0.5359 | 0.3532 | N/A |
| CBM-J | 0.9041 | 0.6171 | 0.7379 | 0.5840 | 0.6017 | 0.5648 | 0.5207 | 0.4922 | 0.4332 | N/A |
| ICIAP-Seq | 0.8014 | 0.6511 | 0.5904 | 0.6160 | 0.4454 | 0.5743 | 0.3625 | 0.5332 | 0.2945 | N/A |
| ICIAP-J | 0.8931 | 0.6016 | 0.7282 | 0.6116 | 0.5851 | 0.5785 | 0.5071 | 0.4957 | 0.4195 | N/A |
| LaBo | 0.9065 | 0.7630 | 0.7190 | 0.6230 | 0.5785 | 0.6955 | 0.4639 | 0.6630 | 0.3009 | N/A |
| **$MC^2$** | **0.9487** | **0.3358** | **0.8450** | **0.3080** | **0.7913** | **0.3177** | **0.7484** | **0.2408** | **0.7022** | N/A |

Table A7: Per Experience Results for CIFAR100

| Model | Experience 1 | | Experience 2 | | Experience 3 | | Experience 4 | | Experience 5 | |
|---|---|---|---|---|---|---|---|---|---|---|
| | AA | Forget | AA | Forget | AA | Forget | AA | Forget | AA | Forget |
| CBM-Seq | 0.6861 | 0.2302 | 0.6067 | 0.1052 | 0.5818 | 0.0947 | 0.5349 | 0.1085 | 0.5329 | N/A |
| CBM-J | 0.7467 | 0.2413 | 0.6794 | 0.1867 | 0.6601 | 0.2270 | 0.5988 | 0.1562 | 0.5874 | N/A |
| ICIAP-Seq | 0.6861 | 0.2302 | 0.6067 | 0.1052 | 0.5818 | 0.0947 | 0.5349 | 0.1085 | 0.5329 | N/A |
| ICIAP-J | 0.7467 | 0.2413 | 0.6794 | 0.1867 | 0.6601 | 0.2270 | 0.5988 | 0.1562 | 0.5874 | N/A |
| LaBo | 0.6483 | 0.4927 | 0.6176 | 0.3907 | 0.5651 | 0.4970 | 0.4244 | 0.5160 | 0.3101 | N/A |
| **$MC^2$** | **0.8615** | **0.0840** | **0.8536** | **0.0433** | **0.8487** | **0.0592** | **0.8119** | **0.0396** | **0.8137** | N/A |

Table A8: Per Experience Results for CUB (Caltech-UCSD Birds-200-2011)

| Model | Experience 1 | | Experience 2 | | Experience 3 | | Experience 4 | | Experience 5 | |
|---|---|---|---|---|---|---|---|---|---|---|
| | AA | Forget | AA | Forget | AA | Forget | AA | Forget | AA | Forget |
| CBM-Seq | 0.8760 | 0.5439 | 0.6650 | 0.4835 | 0.5106 | 0.4400 | 0.3853 | 0.3565 | 0.3967 | N/A |
| CBM-J | 0.8907 | 0.4546 | 0.7300 | 0.4261 | 0.6099 | 0.4175 | 0.4697 | 0.3106 | 0.4961 | N/A |
| ICIAP-Seq | 0.8076 | 0.4950 | 0.5999 | 0.4282 | 0.4587 | 0.4548 | 0.3438 | 0.4016 | 0.3554 | N/A |
| ICIAP-J | 0.8946 | 0.4732 | 0.7154 | 0.4218 | 0.6041 | 0.4410 | 0.4728 | 0.2865 | 0.4868 | N/A |
| LaBo | 0.5720 | 0.5083 | 0.4140 | 0.3599 | 0.4379 | 0.2559 | 0.4244 | 0.2059 | 0.3652 | N/A |
| **$MC^2$** | **0.9351** | **0.0919** | **0.8731** | **0.1228** | **0.8133** | **0.1109** | **0.7541** | **0.0268** | **0.7985** | N/A |

Table A9: Per Experience Results for ImageNet100

### A1.4 MORE QUALITATIVE RESULTS

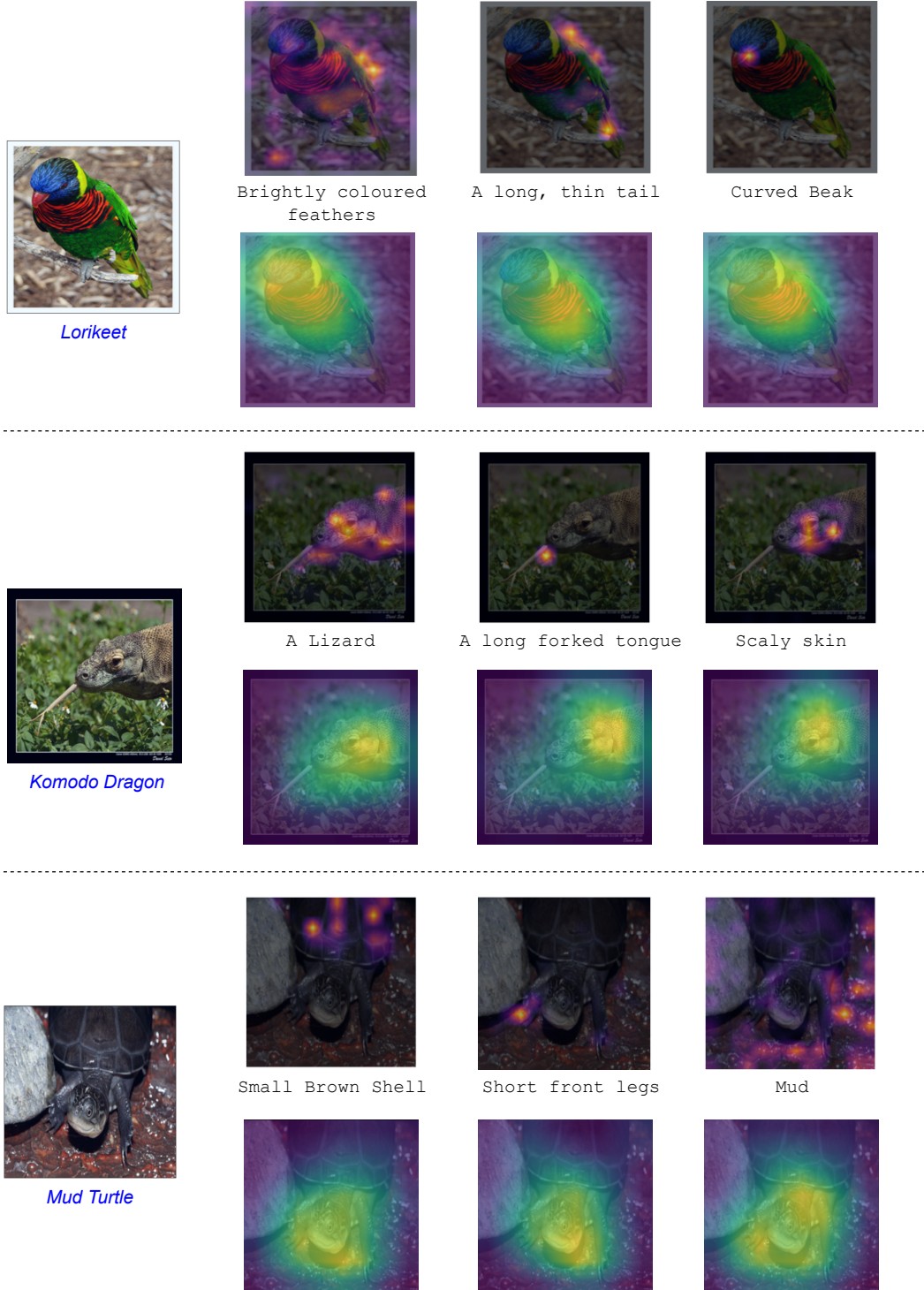

Figure A5: More qualitative results. Alternate rows present the localization results using $MC^2$ versus when localizing the same concepts using GradCam on CBMs

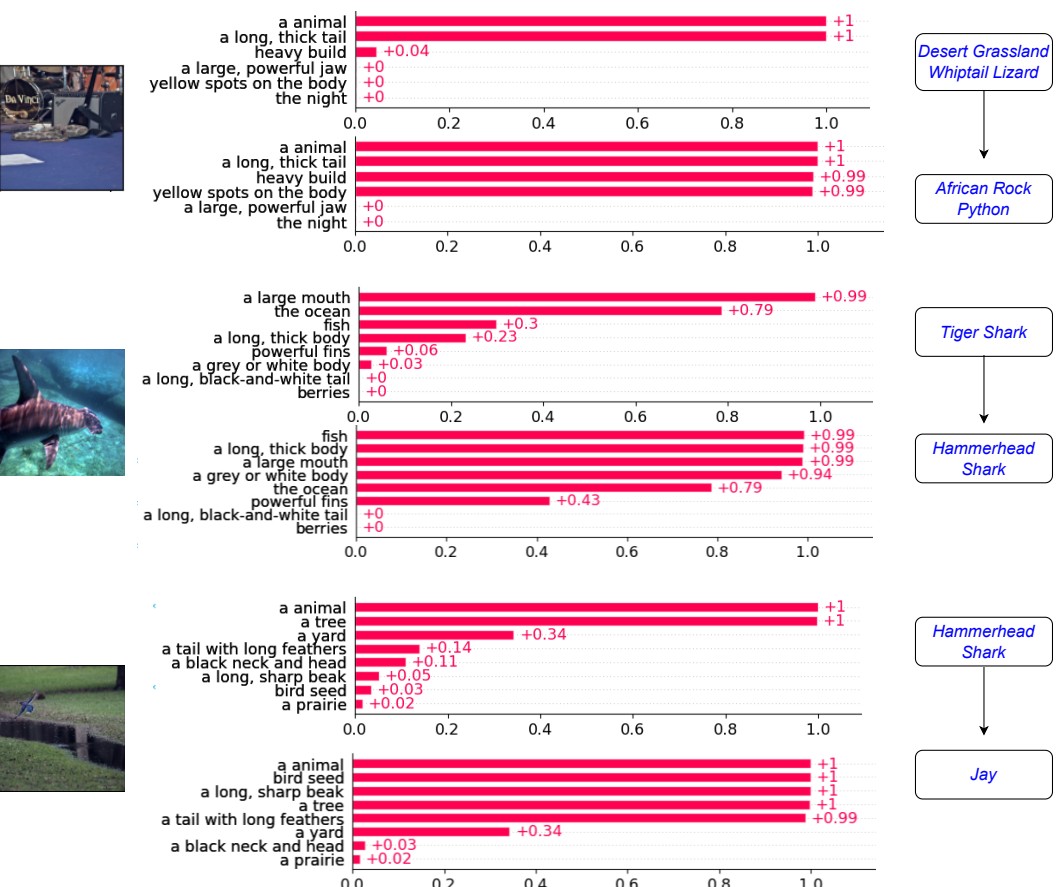

Figure A6: More results on manual interventions

| Dataset | Class | Concepts |
|---|---|---|
| CIFAR100 | Bicycle | a tire, object, a helmet, a handlebar, a bicycle seat, pedals attached to the frame, mode of transportation, two wheels of equal size, a seat affixed to the frame, a chain |
| | Chair | furniture, a person, object, legs to support the seat, an office, a computer, a desk, four legs, a backrest, armrests on either side |
| | Kangaroo | a grassland, short front legs, an animal, a safari, mammal, a long, powerful tail, brown or gray fur, marsupial, long, powerful hind legs, Australia |
| Imagenet100 | Chickadee | trees, grayish upperparts, vertebrate, a short beak, white cheeks, chordate, gray wings and back, an animal, leaves, a small, round shape |
| | American Bullfrog | an animal, a stream, a large size, a log, a webbed foot, a marsh, a lily pad, long, powerful hind legs, a large body, a swamp, a carnivorous diet, a lake, a woods, large, webbed hind feet, a large mouth, a river, a pond, spots or blotches on the skin, a green or brown body |
| | Komodo Dragon | a large size, a keeper, scales, a tree, a dish, scaly skin, a rock, long, sharp claws, a long, thick tail, a long, forked tongue, an animal, reptile, a fence, vertebrate, a water dish, a zoo, a heat lamp, a large, bulky body, a cage, a lizard |
| CUB | Black-footed Albatross | back pattern: solid, under tail color: rufous, wing shape: long-wings, belly color: red, wing color: red, upperparts color: brown, breast pattern: multi-colored, upperparts color: rufous, bill shape: cone, tail shape: notched tail, back color: blue |
| | American Crow | back pattern: solid, wing shape: long-wings, upperparts color: brown, bill shape: cone, tail shape: notched tail, back color: blue, under tail color: grey, wing shape: tapered-wings, belly color: iridescent, wing color: iridescent |
| | Lazuli Bunting | back pattern: solid, under tail color: rufous, throat color: pink, wing shape: long-wings, wing color: red, upper tail color: pink, upperparts color: brown, breast pattern: multi-colored, bill shape: cone, tail shape: notched tail |

Table A10: Sample classes and a subset of their corresponding concepts for the three datasets

