# OpenReview forum: "$MC^2$: Multimodal Concept-based Continual learning"
_ICLR.cc/2024/Conference — ICLR 2024 Conference Withdrawn Submission_

### Official Review · Reviewer_U4mx · 2023-10-28

**Soundness:** 2 fair
**Presentation:** 2 fair
**Contribution:** 2 fair
**Rating:** 3
**Confidence:** 4

**Summary:**

This paper proposes a concept-learning based approach for continual learning. The idea is to learn concepts and categories jointly by referring to a replay buffer. Concepts are represented as encoded features from text descriptions. The method builds upon concept bottleneck methods and learns a multimodal encoder whose outputs can be used to compute similarities with concept representations and make predictions of categories.

**Strengths:**

The method is simple: loss function contains two parts and each part is easy to understand; the strategy to prevent forgetting is through a replay buffer that stores all past exemplars.

The experiments to evaluate concept learning are sufficient.

**Weaknesses:**

1. The novelty of the proposed method is limited. The method targets for interpretable continual learning whereas the only technique is to create a replay buffer and replays exemplars during training. This method is similar to most replay methods in continual learning, such as GSS, iCaRL.

2. Experiments are not convincing for interpretable continual learning. Except for results in Tab. 2, all other results are about ablating the concept learning. Since the main goal of this paper is to evaluate under interpretable continual learning, the authors should design specific experiments to show how different classes are forgotten or strengthened through their approach and how the forgetting is mitigated by training with concepts and it might be beneficial to investigate what concepts are useful to category memorization and what are not. The current experiments do not support a good design for continual learning.

3. No clear measurements for the effect of manual intervention. Besides, the examples in Fig. 4 do not make enough sense to me. Since a tree and a dish are obviously incorrect (concept confidences are near 100% but the concepts do not exist in the image), wouldn’t it be more straightforward to make interventions over these concepts?

4. Technical details are missing. For example, I failed to find whether the exemplars in the replay buffer are all past exemplars or the buffer has a limit.

**Questions:**

See the 3 and 4 in the weaknesses.

---

### Official Review · Reviewer_pP9w · 2023-10-31

**Soundness:** 1 poor
**Presentation:** 2 fair
**Contribution:** 3 good
**Rating:** 3
**Confidence:** 5

**Summary:**

The paper addresses the under-explored issue of neural network interpretability in a continual learning context, where models may suffer from performance degradation due to catastrophic forgetting of previously learned tasks and insufficient human interpretability of the learned representations. To bridge this gap in interpretability, the authors adopt emerging concept-based learning approaches. They hypothesize that generating multimodal concept vectors by combining text encodings of high-level image concepts with image encodings can implicitly localize text-based concepts within images and provide comprehensive interpretability.

Their proposed approach, MC^2, introduces a new continual learning paradigm through concept-augmented replay, enabling models to retain concept-based explanations of prior experiences. They evaluate MC^2 on CUB200, CIFAR100, and ImageNet100 datasets, leveraging Large Language Models (LLMs) to generate text encodings for natural language concepts and image labels. A multimodal encoder facilitates communication between multimodal vectors, facilitating information sharing across modalities. By employing concept grounding, the authors aim to align output vectors with human-interpretable text concepts.

MC^2 represents a novel multimodal concept-based continual learning methodology aimed at improving the interpretability of continually learned methods in safety-critical scenarios. It incorporates text-based concept labels, input images, and class labels into the continual multimodal model. Experience replay, now including concept labels, is utilized to mitigate catastrophic forgetting. A concept-grounding module aligns multimodal inputs with textual concept descriptions, enabling the implicit attribution of concept labels to image regions. The authors demonstrate that this approach enhances both multimodal performance and interpretability. MC^2 outperforms concept-based methods in class-incremental settings on ImageNet-100, CIFAR-100, and CUB-200, while maintaining comparable performance in the full data setting, all without requiring additional parameters.

**Strengths:**

Introduces a novel approach to addressing catastrophic forgetting and model interpretability jointly.

Explores the use of concepts-learning as a proxy for interpretability.

Proposes the use of LLMs to avoid costly dense concept annotation, although this is only used for some datasets and not analyzed in terms of how well it works.

Suggests the use of a weighted binary cross-entropy loss to handle a small ratio of active concepts to total concepts.

Provides qualitative results showing improved class activation maps (CAM) compared to the CBM counterpart.

Offers scalability through the use of linear attention variants for practical implementation.

Conducts vision-alignment ablations to analyze the methodology further.

Highlights the potential reusability of concepts for downstream tasks, contributing to the continual learning problem's improvement.

**Weaknesses:**

More detailed analysis of attribution maps in the continual learning setting, including examples where concept attribution quality deteriorates compared to an oracle baseline, would enhance the paper's assessment of interpretability. Investigate when this deterioration occurs in relation to standard continual learning metrics to make the approach more practical.

I do not know why the need to create concepts with an LLM is necessary for CUB-200. For CUB-200, each image has detailed annotations: 1 subcategory label, 15 part locations, 312 binary attributes and 1 bounding box. One could quantitatively compare the LLM annotations to those from CUB-200.

The datasets studied are really not sufficient for 2023. People have been studying ImageNet-1K in continual learning since at least 2016 and we have far more compute nowadays. Many methods do not scale.

Additional analysis is needed to showcase the strengths over conventional replay to replay with concepts. Specifically, is the system more sample efficient or does it achieve other benefits over standard experience replay? Moreover, the comparison methods are inadequate and given the focus on replay, the authors need to compare against strong replay based methods, e.g., DerPP, Experience Replay, REMIND, etc. I think Table 2 is misleading given this. I really think the choice of comparison models should be focused on other replay methods vs. growing networks or adding in more parameters, etc.

Earlier work on multi-modal continual learning should be cited and discussed in related work, e.g., https://www.ecva.net/papers/eccv_2020/papers_ECCV/papers/123530460.pdf

Compare and contrast the proposed concept grounding module with previous concept-based methods to establish its novelty more explicitly.

Include a discussion of the caption generation process using Large Language Models (LLMs) for ImageNet-100 and CIFAR-100.

Analyze how the quality and type of generated captions impact performance and interpretability.

Explore the scalability of compute requirements when using LLMs for large datasets with more capable models.

Address the lack of quantitative results or analysis on the learning and retention of new concepts, as well as the reuse of older concepts for new ones.

There is a need to develop new evaluation methods and benchmarks to provide a more holistic evaluation of the proposed approach.

Provide information on how concepts are distributed across several experiences and whether efforts have been made to address issues like concept leakage.

Discuss how concept distribution aligns with class distribution in conventional datasets and address any efforts to balance these distributions.


There seems to be no specific mention of the caption generation process  using LLMs for ImageNet-100 and CIFAR-100. It would be important to understand how the quality/type of generated captioning affects performance/interpretability, as well as the
scalability of compute requirements for a large dataset using a sufficiently capable LLM for this task.

Consider comparing pretrained and randomly initialized encoders for training to observe the effect of text/image alignment with pretrained and tabula-rasa models. The authors are throwing very powerful pre-trained networks at relatively small datasets where the models have already seen data that likely comes from the distributions studied.

Offline upper bounds need to be presented to calibrate how effective the method is.

Multiple orderings should be studied, as in real-world continual learning one would not know what the nature is for the distribution to be learned.

Consider using this dataset which is a large-scale dataset with attributes annotated: https://openaccess.thecvf.com/content/CVPR2021/html/Pham_Learning_To_Predict_Visual_Attributes_in_the_Wild_CVPR_2021_paper.html

Overall, I like the general framework, but I think there are a number of confounds that need to be addressed, where baselines are inadequate and the experiments are not sufficient for assessing performance. Given the use of powerful pre-trained networks, I'd expect the experiments to likewise be designed for large-scale problems, where the author's demonstrate multiple advantages for their method over others in terms of increased interpretability and superiority of concept-based replay over conventional replay. Small toy datasets do not suffice for 2023, especially when using large pre-trained networks. I assume the ViT was already pre-trained on ImageNet-1K or ImageNet-21K, so the experiments are very confounded for ImageNet-1K and the other datasets in that the images are not really out-of-distribution.

**Questions:**

Elaboration required on types of concepts. Authors mention “high level semantics” associated with object category, which seems far too general. What, if any, should be the restrictions to or types of the class concepts generated, given that an LLM is employed for the caption generation?

How would the strength scores’ distribution vary over time as  more concepts will be introduced? Would that lead to “undecisive” models whose strength score distribution might get flatter?

What are other ways than concept-based learning might be a stronger implication of  interpretability? In your discussion, you point out some of the past work using part-based prototypes and you choosing not to go in this direction. What was the rationale for this?

How would you measure the usefulness, transferability, and reusability of the concepts over longer periods of time in terms of the number of tasks?

---

### Official Review · Reviewer_49aA · 2023-11-04

**Soundness:** 3 good
**Presentation:** 2 fair
**Contribution:** 2 fair
**Rating:** 5
**Confidence:** 3

**Summary:**

This paper proposes MC2, a new framework that focuses on the perspective of concept-based models that classes are considered
as combinations of text-based concepts. The MC2 could enhance the interpretability of models under the continual learning setting, and solve the challenge of forgetting old knowledge when learn new ones. Experimental result prove the effectiveness of this proposed approach.

**Strengths:**

1. The idea of learning concept for different incoming class is a novel perspective for solving forgetting and interpretability simultaneously, especially the concept-grounding module, which could ground various multimodal concept vectors to the existing text description. I think this design could also used for open concept description and recognization.

2. The description for the experimental setting and result seems well, table 2 shows the very promising performance from MC2. Following discussion and ablation study also convinced.

**Weaknesses:**

1. I think the assumption of the MC2 maybe a little strong, majorly from the following perspectives:
a. the input tuples need the available concepts for each input instance-label. Although the author propose the solution for collecting concept annotations like LLM query, it may not effective in specific domain (LLM description for the CIFAR and ImageNet datasets with general concept is enough). What's the performance if weak annotation, or what's the precise degree for concept that model depends on;

b. some concepts may share common feature/concept annotation tokens, e.g., fur, four legs, pointy ears maybe suitable for both cat and dog. The author did not describe clearly how to learn class-agnostic and class-specific concept. Also, what would be happened if missing modality happened? (concept may missing, or text is missed)

2. Experimental result need some further result to be specific convinced, e.g., the experience seems similar with task in previous continual learning work. The paper only show 5-experience on CUB,CIFAR and ImageNet, is it possible to extend to 10, 20 experiences? Also, the author apply the experience replay during the model adaptation, what's the influence of the amount for storing data?

**Questions:**

For storing old concept as experience replay, is it possible to use prompt here to represent the concept?

---

### Official Review · Reviewer_Nstq · 2023-11-06

**Soundness:** 2 fair
**Presentation:** 1 poor
**Contribution:** 2 fair
**Rating:** 3
**Confidence:** 4

**Summary:**

The paper introduces a vision-language concept-based continual learning approach. The paper follows the conventional continual learning benchmarks like CIFAR-100 and ImageNet-100, yet additionally provides concept sets in the training phase. Here, a data instance in each task contains multiple part-based/text-based semantics describing the data, and they can overlap different tasks/classes over time. In view of the architecture, they introduce a single-layered feedforward network called a concept grounding module that computes the similarity of concept embeddings and the projected multimodal concept vector. They have compared the proposed method to multiple concept-based learning methods and achieved superior performance.

**Strengths:**

- Suggest a new multimodal approach for interpretable/concept-based continual learning.
- Provide multiple quantitative/qualitative analyses for concept-based continual learning.

**Weaknesses:**

- The claim is not strong. In the introduction, the authors argue that there are several new problems in concept-based continual learning against conventional continual learning scenarios. However, the first and second challenges - (i) learning new concepts without forgetting (ii) addressing catastrophic forgetting of concepts while learning new classes highly duplicate each other, and hard to say these are 'new challenges'. Simply adopting continual learning scenarios on concept-based deep learning is a bit incremental. I strongly recommend that the authors study the impact of learning new concepts incrementally during continual learning scenarios beyond a simple combination of them.

- Additionally, the third challenge, (iii), describes that earlier concepts can crucially affect later tasks. This could be an exciting part against conventional continual learning settings that are barely interpretable. However, throughout the paper, this challenge is not analyzed. Also, the paper emphasizes that they focus on generic concept-based incremental learning, including both part-based and high-level semantic concepts, unlike previous work (Rymarczyk et al., 2023). However, detailed differences between the two concepts are not studied in the paper, and if I'm correct, all provided examples in Figures 1, 2, 3, and 4 are simple part-based concepts.

- Task is limited to image classification. Concept-based learning is more promising on diverse tasks, such as Visual Question Answering, Attribute Learning, and Multimodal Generation, but the suggested problem setting is a simple combined version of two problems, which is less attractive.

- In my understanding, this training framework requires oracle concept labels for all continual learning tasks in advance. There are unlimited available concepts and classifying them with a parametric classifier is infeasible in real-world situations.

- Notations and presentations can be further improved. Multiple similar notations (e.g., $x$ and **x**, 'language encoder' and 'text encoder', a set of concepts $\mathcal{C}\_{i}$, $|\mathcal{C}^{i}_{NL}|$, and concept embedding **c**, etc.) with ambiguous and insufficient descriptions harm readability of the paper.

- Baselines are too naive - just increment classifiers/modules without continual learning techniques. They obviously fail this problem by design since they do not contain some regularization/replay buffer approaches, widely used in continual learning and related application literature. Therefore, the current quantitative evaluation can not demonstrate the effect of suggested methods in continual learning scenarios. At least, the baseline should be combined with recent strong continual learning strategies. A comparison with Rymarczyk et al., 2023 is also recommended.

- Lack of critical analyses on the concept-based continual learning. No study on concept forgetting.  There could be a number of informative, in-depth discussions, such as the impact of concept forgetting on task differences and concept duplication rates, etc.

**Questions:**

.